# Beeswax Alcohol Prevents Low-Density Lipoprotein Oxidation and Demonstrates Antioxidant Activities in Zebrafish Embryos and Human Subjects: A Clinical Study

Kyung-Hyun Cho [1,*], Seung-Hee Baek [1], Hyo-Seon Nam [1], Ashutosh Bahuguna [1], Luis Ernesto López-González [2], Iván Rodríguez-Cortina [2], José Illnait-Ferrer [2], Julio César Fernández-Travieso [2], Vivian Molina-Cuevas [2], Yohani Pérez-Guerra [2], Ambar Oyarzabal Yera [2] and Sarahi Mendoza-Castaño [2]

[1] Raydel Research Institute, Medical Innovation Complex, Daegu 41061, Republic of Korea
[2] National Centre for Scientific Research, Havana 11300, Cuba
[*] Correspondence: chok@raydel.co.kr; Tel.: +82-53-964-1990; Fax: +82-53-965-1992

**Abstract:** Oxidative stress is one of the primary instigators of the onset of various human ailments, including cancers, cardiovascular diseases, and dementia. Particularly, oxidative stress severely affects low-density lipid & protein (LDL) oxidation, leading to several detrimental health effects. Therefore, in this study, the effect of beeswax alcohol (BWA) was evaluated in the prevention of LDL oxidation, enhancement of paraoxonase 1 (PON-1) activity of high-density lipid & protein (HDL), and zebrafish embryo survivability. Furthermore, the implication of BWA consumption on the oxidative plasma variables was assessed by a preliminary clinical study on middle-aged and older human subjects ($n = 50$). Results support BWA augmentation of PON-1 activity in a dose-dependent manner (10–30 μM), which was significantly better than the effect exerted by coenzyme $Q_{10}$ ($CoQ_{10}$). Moreover, BWA significantly curtails LDL/apo-B oxidation evoked by $CuSO_4$ (final 0.5 μM) and a causes a marked reduction in lipid peroxidation in LDL. The transmission electron microscopy (TEM) analysis revealed a healing effect of BWA towards the restoration of LDL morphology and size impaired by the exposure of $Cu^{2+}$ ions (final 0.5 μM). Additionally, BWA counters the toxicity induced by carboxymethyllysine (CML, 500 ng) and rescues zebrafish embryos from development deformities and apoptotic cell death. A completely randomized, double-blinded, placebo-controlled preliminary clinical study on middle- and older-aged human subjects ($n = 50$) showed that 12 weeks of BWA (100 mg/day) supplementation efficiently diminished serum malondialdehyde (MDA) and total hydroperoxides and enhanced total antioxidant status by 25%, 27%, and 22%, respectively, compared to the placebo-control and baseline values. Furthermore, the consumption of BWA did not exhibit any noteworthy changes in physical variables, lipid profile, glucose levels, and biomarkers pertinent to kidney and liver function, thus confirming the safety of BWA for consumption. Conclusively, in vitro, BWA prevents LDL oxidation, enhances PON-1 activity in HDL, and positively influences oxidative variables in human subjects.

**Keywords:** beeswax alcohol; clinical trials; high-density lipoproteins; low-density lipoproteins; oxidative stress; paraoxonase-1; zebrafish

## 1. Introduction

Free radicals are highly reactive molecular species with several detrimental effects on the living being. Reactive oxygen species (ROS) such as hydroxyl radicals ($^{\bullet}OH$), superoxide anions ($O_2^{\bullet-}$), singlet oxygen ($^1O_2$), and hydrogen peroxide ($H_2O_2$) are the most common subtypes of free radicals that are consistently generated in the living organisms via various metabolic processes and are countered by endogenous enzymatic and nonenzymatic antioxidant defense systems that maintain cellular homeostasis [1]. However, several internal and external factors such as age, disease, pollution, smoking, alcohol, radiation,

and consumption of certain drugs (like cyclosporine, bleomycin, and gentamycin) are the causative agents for the excessive cellular production of reactive oxygen species (ROS) that is beyond the threshold limit of the endogenous antioxidants, which leads to oxidative stress [1]. ROS-induced oxidative stress has destructive ramifications on all classes of biomolecules, mainly lipids, proteins, and DNA [2], and has been accredited as a direct or indirect reason behind more than 100 types of human ailments [3,4]. The intimate involvement of oxidative stress to stimulate different kinases-incurring pathways and transcription factors like nuclear factor kappa B (NFκB) and activator protein 1 (AP-1) suggests the deep impact of oxidative stress on the induction of inflammation and inflammation-related disorders [5,6].

Several clinical and preclinical studies implicated oxidative stress and its induced effects as a major culprit for the onset of several neurological, kidney, and respiratory disorders [1]. A notable effect of oxidative stress is well established for hypertension, ischemia, atherosclerosis, cardiomyopathy, cardiac hypertrophy, and other cardiovascular diseases. Oxidative stress has also been shown to impose a significant impact on circulating low-density lipid & protein (LDL), leading to LDL oxidation (oxLDL) [1]. The oxLDL are easily recognized/probed by scavenger receptors on macrophages, which spurs foam-cell generation and leads to lipid accumulation and atherosclerotic plaque formation [7]. Interestingly, high-density lipid & protein (HDL) inhibit LDL oxidation and greatly impact atherosclerosis [7]. HDL, owing to their antioxidant activity attributed by enzymes such as paraoxonase 1 (PON-1), lecithin-cholesterol acyltransferase (LACT), and platelet-activating factor acetyl hydrolase (PAF-AH), inhibit and prevent the LDL oxidation [8] and the consequent impact on atherosclerosis. Nevertheless, lipid and protein constituents of HDL are also susceptible to oxidative alterations [9], impairing HDL functionality and thereby increasing the risk of atherosclerosis and various other diseases.

Knowing the numerous detrimental effects of oxidative stress, its management is imperative, which can be accomplished by supplementation with external antioxidants that work indigenously or in conjunction with the endogenous antioxidant system for the effective elimination of surplus ROS. Several synthetic and natural compounds of plant and animal origin have been documented for their potential antioxidant activities [10,11] using different in vitro and in vivo assays. However, a limited number of them have undergone human testing to demonstrate their efficacy as antioxidants for human consumption.

Beeswax alcohol (BWA) is a blend of six linear aliphatic alcohols (of carbon chain C24 to C34) extracted from beeswax [12]. Owing to its hydrophobic surface and insolubility in water, BWA exhibits resistance to digestive enzymes; however, the solubility of BWA is enhanced in the presence of bile acid [13]. The investigation of BWA absorption, distribution, metabolism, and excretion is challenging due to its intricately complex composition [13]. Recently, we successfully synthesized reconstituted high-density lipoproteins (rHDL) in vitro, incorporating both apoA-I and BWA. Notably, BWA demonstrated strong binding affinity with apoA-I, facilitating the formation of rHDL at a molar ratio of 1:0.5 and 1:1 for apoA-I to BWA [14]. ApoA-I, primarily expressed in the intestine and liver, enters the circulation in association with chylomicrons and is swiftly transferred to HDL. As hepatic apoA-I is released into the circulation during the conversion of chylomicrons into chylomicron remnants by lipoprotein lipase [15,16], it is present in a lipid-free or minimally lipidated form [15,17]. This observation suggests that BWA may be absorbed in the small intestine through a potential binding and interaction with apoA-I. BWA is well recognized for its diverse biofunctionality, including antioxidant, antiplatelet, and anti-inflammatory activity, by inhibiting cyclooxygenase (COX) and 5-lipoxygenase (5-LOX) [12,18]. However, the most noteworthy role of BWA was established as a gastroprotective agent. Several pharmacological and clinical studies confirmed the impact of BWA on the quantity and quality of gastric mucous [19,20]. The Korean Food and Drug Administration (KFDA) recently approved BWA as a functional food ingredient for improving joint and gastrointestinal health (https://www.foodsafetykorea.go.kr/portal/healthyfoodlife/searchHomeHFDetail.do?prdlstReportLedgNo=2023021000330706 (accessed on 16 May 2023)). Despite the several

health-beneficial effects, BWA has not been tested regarding its impact on paraoxonase-1 (PON-1) activity or as a preventive agent against LDL and HDL oxidative damage.

Taking these points into consideration, the present study aimed to evaluate the effect of BWA on HDL-associated PON-1 activity and the prevention of LDL and HDL oxidative damage. Additionally, the protective effect of BWA was tested in zebrafish embryos against carboxymethyllysine (CML)-induced stress. Finally, BWA (100 mg/day) was processed for the preliminary clinical trials to assess the effect of BWA on the oxidative variables, physical variables, liver and kidney function biomarkers, and lipid profile of middle-aged and older human subjects of both genders.

## 2. Materials and Methods

### 2.1. Materials

Beeswax alcohol (BWA) sourced from the National Center for Scientific Research (CNIC), Havana, Cuba, was complimentary from Raydel Australia, Pty, Ltd. (Thornleigh, NSW, Australia). The BWA was specified with six high-molecular-weight alcohols (i.e., triacontanol ($C_{30}H_{62}O$) (25–35%), dotriacontanol ($C_{32}H_{66}O$) (18–25%), octacosanol ($C_{28}H_{58}O$) (12–20%), hexacosanol ($C_{26}H_{54}O$) (1–20%), tetracosanol ($C_{24}H_{50}O$) (6–15%) and tetratriacontanol ($C_{34}H_{70}O$) ($\leq$7.5%) extracted from beeswax (*Apis mellifera* L.) [12]. All other chemicals unless otherwise stated were of the highest purity and used as supplied. 2,2-Diphenyl-1-picrylhydrazyl (Cat. No. 044150), paraoxon-ethyl (Cat. No. 36186), *N-ε*-carboxymethyllysine (Cat No. 14580-5 g), coenzyme $Q_{10}$ ($CoQ_{10}$) (Cat. No. C9538-100 mg), dihydroethidium (Cat. No. 37291), and acridine orange (Cat. No. A9231) were purchased from Sigma-Aldrich (St. Louis, MO, USA).

### 2.2. DPPH Free Radical Scavenging Activity

The antioxidant activity of BWA and $CoQ_{10}$ was determined by 2,2-diphenyl-1-picrylhydrazyl (DPPH) free radical assay as in an earlier adopted method [21]. In brief, 950 μL DPPH solution (24 μg/mL in methanol) was mixed with 50 μL of BWA or $CoQ_{10}$ to obtain the final concentrations of 0.125, 0.25, 0.5, 2.5, and 5 μM. Simultaneously, DPPH solution devoid of BWA and $CoQ_{10}$ was prepared that served as the control. DPPH radical scavenging activity was measured after 120 min incubation at 25 °C by taking absorbance at 517 nm using a spectrophotometer (UV-2600i-Shimadzu, Kyoto, Japan).

### 2.3. Isolation of LDL and HDL

LDL and HDL from human blood were isolated using density-gradient centrifugation, as in the previously described method [22]. The blood was donated voluntarily by the human participants after 12 h of fasting and collected according to Helsinki guidelines approved by the Institutional Review Board of Korea National Institute for Bioethics Policy (KoNIBP, approval number P01-202109-31-009), supported by the Ministry of Health Care and Welfare (MOHW) of Korea. For the isolation of LDL (1.019 < d < 1.063) and HDL (1.063 < d < 1.225), the density of the gradient was adjusted with NaCl and NaBr using the standard method [23], and the content was centrifuged at 100,000× *g*. After 24 h centrifugation, LDL and HDL were recovered from the respective density fractions and dialyzed overnight against Tris-buffered saline (pH 8.0) to remove the trace of NaBr. The dialyzed LDL and HDL were preserved in the refrigerator for further use.

### 2.4. Paraoxonase Assay

Paraoxonase-1 (PON-1) activity was determined using HDL following the previously described method [24,25]. In brief, HDL (2 mg/mL, 20 μL) was mixed individually with varying concentrations (0.01–2.5 μM) of BWA or $CoQ_{10}$. Subsequently, the substrate paraoxon-ethyl (550 mM) was added to the reaction mixture, and the content was incubated at 25 °C. After 215 min incubation, an absorbance of 415 nm was recorded to quantify p-nitrophenol production, a hydrolysis product of paraoxon-ethyl catalyzed by PON-1. The PON-1 activity is expressed as μU/L/min, and μU is defined as the production of

1 μmol/L p-nitrophenol by unit volume of the enzyme at the unit time considering the $17 \times 10^3$ $M^{-1}cm^{-1}$ molar absorption coefficient for p-nitrophenol.

## 2.5. Oxidation of LDL, Electrophoresis, and Quantification of LDL-Oxidized Products

The $CuSO_4$-induced oxidation of LDL and the effect of BWA and $CoQ_{10}$ on LDL oxidation was determined by the previously described method [26]. In brief, LDL (final 1 mg/mL, 75 μL) was mixed with $CuSO_4$ (final 0.5 μM, 7.5 μL) with the subsequent addition of BWA or $CoQ_{10}$ to achieve a final concentration of 10, 20, and 30 μM. After 30 min incubation at 37 °C, the content was electrophoresed on 0.5% agarose gel (nondenaturing conditions) using Tris-EDTA running buffer (pH 8.0) at a constant electric supply of 50 V. After about an hour, electrophoresis was terminated, and the separated proteins LDL/apo-B were visualized by Coomassie blue staining (final 1.25%). The degree of oxidation in LDL was quantified by thiobarbituric acid reactive substance (TBARS) assay using malondialdehyde (MDA) as the standard following the earlier described method [22,27].

## 2.6. Electron Microscopic Examination of LDL

The transmission electron microscopic (TEM) examination of LDL and oxidized LDL (oxLDL) treated with BWA (10–30 μM) was performed as in the previously described method [22] with slight modification. In brief, an equal amount (5 μL) of LDL suspension, which was stained with 2% phosphotungstate acid (pH 7.4), was placed over carbon-coated 200-mesh copper grids. After 2 min, the excess content was blotted, and the LDL-attached copper grid was incubated for 2 h at 50 °C. Finally, LDL particle morphology was observed under TEM (Hitachi H-7800; Ibaraki, Japan) at 80 kV acceleration voltage.

## 2.7. Exposure of HDL to Ferrous Ions

The effect of BWA to prevent the ferrous ion ($Fe^{2+}$)-mediated oxidative damage of HDL was examined by SDS-polyacrylamide gel electrophoresis (PAGE) as in the previously described method [28]. In brief, HDL (2 mg/mL) was mixed with ferrous sulfate ($FeSO_4$) (60 μM final), and subsequently, different amounts of BWA (0, 10, 20, and 30 μM) were added. The content was incubated at 37 °C for 72 h in the presence of 5% $CO_2$ following 15% PAGE. The separated bands were stained with commissive brilliant blue (final 1.25%) to document the stability of apoA-I in HDL. The ferrous ion-mediated damage was examined by quantifying the band intensity using Image J software (http://rsb.info.nih.gov/ij, 1.53r version, accessed on 16 January 2023). Further, the morphological analysis of HDL was examined using TEM analysis following the methodology outlined in Section 2.6.

## 2.8. Microinjection of Zebrafish Embryos

Zebrafish embryos were produced as in the previously described method [29]. In brief, zebrafish (of both sexes) were segregated from each other for 16 h, followed by coupling of males and females (2:1) in the breeding tank to produce embryos. The produced embryos (1.5 h postfertilization) were randomly divided into four groups (each group $n = 150$). In Groups I and II, embryos were microinjected with 10 nL PBS (vehicle) and 10 nL PBS containing 500 ng CML. The embryos in Groups III and IV were microinjected with 500 ng CML together with 10 nL of $CoQ_{10}$ (final 15 μM) or BWA (final 15 μM), respectively. Embryos were constantly monitored under the microscope (Motic, Motic SMZ 168; Kowloon Bay, Kowloon, Hong Kong, China) to examine their developmental deformities and survivability, and the images were captured at 5 h and 24 h post-treatment.

## 2.9. Reactive Oxygen Species (ROS) and Apoptosis in Zebrafish Embryos

Reactive oxygen species (ROS) in embryos were evaluated with dihydroethidium (DHE) fluorescent staining as in a previously described method [22]. In brief, zebrafish embryos ($n = 20$) at 5 h post-treatment were stained with DHE (30 μM). After 30 min incubation in the dark, embryos were rinsed twice with $1 \times$ PBS and visualized under a fluorescent microscope (Nikon Eclipse TE2000, Tokyo, Japan) at the excitation and emission

wavelengths of 588 nm and 605 nm, respectively. The apoptosis in the embryos was examined by acridine orange (AO) fluorescent staining [22]. At 5 h post-treatment, zebrafish embryos ($n$ = 20) from different groups were suspended in 500 μL of AO (5 μg/mL). After 1 h staining, embryos were rinsed two times with $1 \times$ PBS and visualized under a fluorescent microscope (Nikon Eclipse TE2000, Tokyo, Japan) at the excitation and emission wavelength of 502 nm and 525 nm, respectively. Image J software was employed for quantifying the fluorescently stained area.

### 2.10. Clinical Study

#### 2.10.1. Participants

The enrolled human subjects were of both sexes, aged between 40 and 75 years, and recruited voluntarily after signing the informed consent. All the participants were healthy as per their medical history, anthropometric examination, and laboratory tests. Subjects diagnosed with thyroid, hepatic, and renal diseases; diabetes (fasting glucose > 7 mmol/L); hypertension (diastolic pressure $\geq$ 100 mmHg); cancer; the blood indicator values of alanine aminotransferase (ALT) or aspartate aminotransferase (AST) > 55 UI and creatinine > 130 μmol/L; or if they had suffered myocardial infarction, unstable angina, stroke, ischemic transient attacks, major surgery, or disease-related hospitalization within the six months prior to this study were excluded. Subjects consuming antioxidant or vitamin supplements within one month before this study were also excluded.

#### 2.10.2. Study Layout

This study was conducted in a randomized, double-blind, placebo-controlled manner in accordance with the Declaration of Helsinki as well as the recommendation of the World Health Organization and the Cuban regulations on good clinical practice. The study protocol was approved by the independent Clinical Research Ethics Committee of the Surgical Medical Research Center (IRB approval number. 04-19), Havana, Cuba, and registered in the Public Registry of Clinical Trials of Cuba (RPCEC00000401). In total, 55 participants were recruited, with 50 included in the active treatment phase. Five subjects were not included due to high baseline values for cholesterol (one subject), glucose >7 mmol/L (three subjects), and ALT > 55 UI (one subject). The 50 participants (both male and female) were randomly divided into two groups ($n$ = 25 in each group) and started to consume two 50 mg tablets of BWA/day (i.e., 100 mg) or placebo for up to 12 weeks, as documented in Figure 1. The BWA dosage (100 mg/day) was determined by referencing the literature [18], indicating the efficacy of BWA (at 100 mg/day dosage) in providing a curative effect on joint health and gastric mucosa.

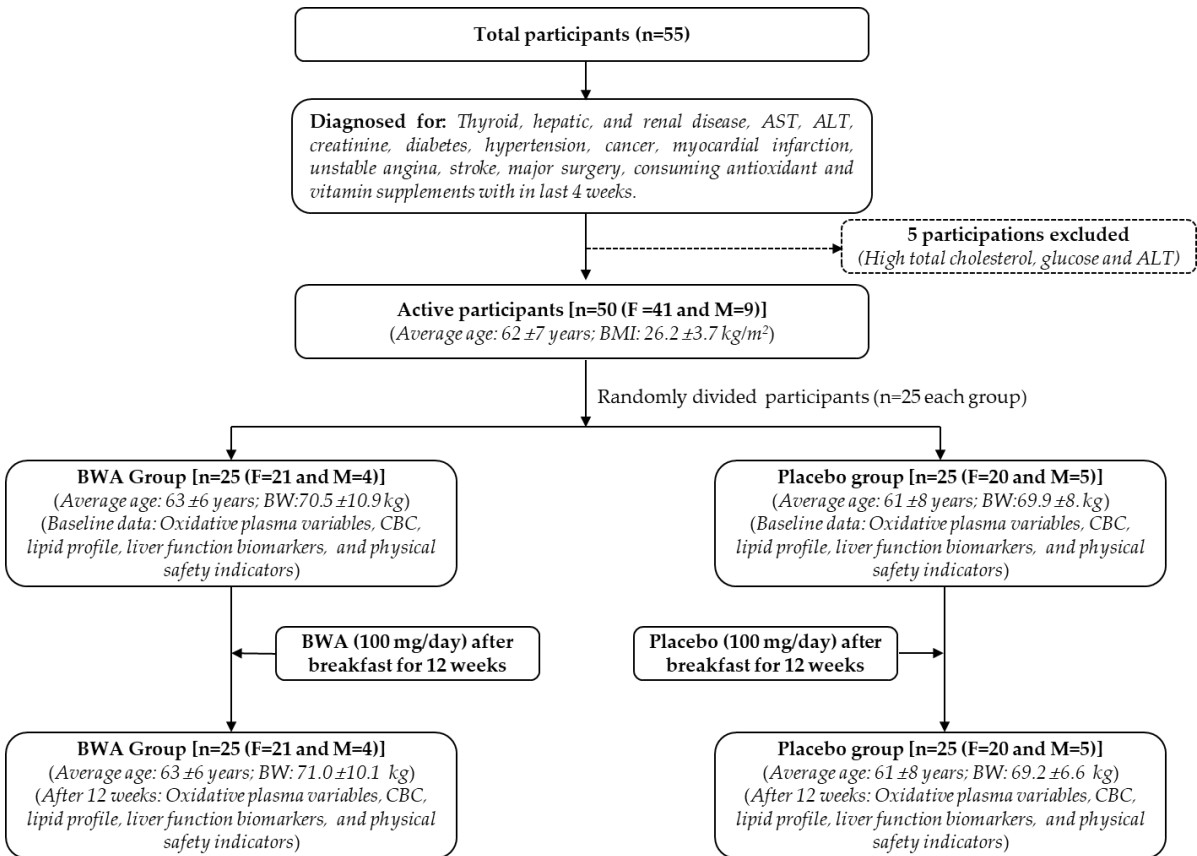

**Figure 1.** Study layout for the clinical trials of beeswax alcohol (BWA). Subjects with ≥100 mmHg (diastolic pressure), >55 UI for ALT and AST, >130 μmol/L creatinine, and >7 mmol/L glucose (fasting) were excluded from this study. AST, ALT, BMI, BW, and CBC are abbreviated for aspartate aminotransferase, alanine transaminase, body mass index, body weight, and complete blood count, respectively. F and M are the abbreviations for female and male, respectively.

### 2.10.3. Anthropometric and Blood Analysis

The participants' body weight, body mass index (BMI), and pulse rate in both groups were analyzed at the beginning (baseline) and after 12 weeks of BWA or placebo consumption. Similarly, the blood was collected from the participants of both groups after overnight fasting in EDTA-rinsed tubes. Blood samples were centrifuged at $3000 \times g$ for 15 min at 4 °C to obtain plasma. Hematological indicators (hemoglobin, hematocrit, red blood cell count, white blood cell count, platelet count) and blood biochemistry indicators (ALT, AST, glucose, creatinine, total cholesterol, and triglycerides) were assessed with enzymatic routine methods using reagent kits (Roche, Basel, Switzerland) in the Hitachi 719 autoanalyzer (Tokyo, Japan) of the Surgical Medical Research Center (Havana, Cuba).

### 2.10.4. Oxidative Variables

In both the groups (BWA or placebo) at baseline and after 12 weeks, lipid peroxidation was assessed through the quantification of plasma MDA using the colorimetric method [30]. Plasma (0.5 mL) was mixed with sodium dodecyl sulfate (SDS) (final 8.1%, 0.2 mL), 20% acetic acid (pH 3.5, 1.5 mL), and an aqueous solution of thiobarbituric acid (TBA) (final 0.8%, 1.5 mL). Finally, the volume was adjusted to 4 mL with distilled water followed by 45 min heating at 95 °C. After cooling, the content was mixed with 1 mL of distilled water and 5 mL of n-butanol: pyridine (15:1 *v/v*) followed by centrifugation at $1500 \times g$ for 10 min. The organic layer was retrieved, and an absorbance of 532 nm was recorded. The concentration of thiobarbituric acid reactive substance (TBARS) was expressed as MDA equivalent/mL using a standard curve of MDA.

Total plasma hydroperoxides level (TOH) was determined by ferrous oxidation in a xylenol orange (FOX) assay as in an earlier described method [31]. In brief, 0.1 mL plasma was mixed with 0.9 mL of FOX reagent and incubated at 37 °C for 30 min. The absorbance of 560 nm was measured, and TOH was quantified using a standard curve of cumene hydroperoxide. The total antioxidant capacity of plasma was evaluated by a colorimetric kit (Randox, Crumlin, UK) that incubated 2,2′-Azido-di-(3-ethylbenzthiazoline sulphonate) (ABTS) with a peroxidase (metmyoglobin) and hydrogen peroxide ($H_2O_2$), which produces a blue-green radical cation ($ABTS^+$) detected at 700 nm.

### 2.11. Statistical Analysis

The outcomes are illustrated as mean ± SD for all the experiments conducted in triplicate. The pairwise statistical differences between the groups were determined through one-way analysis of variance (ANOVA) followed by Tukey's multiple range test, utilizing SPSS software (version 29.0; SPSS Inc., Chicago, IL, USA).

## 3. Results and Discussion

### 3.1. Antioxidant Activity of BWA

The antioxidant activity of BWA was first assessed by DPPH free radical scavenging assay. As depicted in Figure 2A, a 28.8%, 31.4%, 36.8%, and 44.2% reduction in DPPH free radicals was observed at 0.25, 0.5, 2.5, and 5 μM BWA concentrations, respectively, which is significantly better than the activity of $CoQ_{10}$ at the respective concentrations. Precisely at 5 μM concentration, BWA displayed a significant 2.7-fold higher ($p = 0.006$) reduction in DPPH free radicals than that of the $CoQ_{10}$ at the same concentration. These results indicate that BWA can scavenge free radicals and has an antioxidant activity. The results are consistent with the earlier study [32] documenting the DPPH radical scavenging activity of octacosanol, an essential BWA component. Surprisingly, we observed a slightly better activity that can be justified by the presence of different aliphatic alcohols in BWA that work synergistically compared to octacosanol [32]. Additionally, a positive correlation of policosanol content in milk thistle was established for DPPH and 2,2′-azino-bis (3-ethylbenzothiazoline-6-sulfonic acid) (ABTS) radical scavenging activity, corroborating the present findings [33]. Also, the ability of BWA to scavenge endogenous hydroxyl radicals in gastric mucosa supports the BWA antioxidant nature [34].

Further, the effect of BWA was evaluated on the activity of PON-1, an important antioxidant enzyme associated with HDL [35,36]. Therefore, any substance that can enhance PON-1 activity has a direct impact on the functionality of HDL. Knowing this, BWA was tested for its impact on PON-1 activity, and the outcomes were compared with the influence of $CoQ_{10}$ (an endogenous antioxidant in humans and other animals) on the PON-1 activity. The current results demonstrate a noteworthy enhancement of PON-1 activity in the presence of BWA (Figure 2B). A 163.2 μU/L/min PON-1 activity was noticed in the control (without BWA or $CoQ_{10}$) that enhanced in the presence of BWA (0.01–2.5 μM) and attained a maximum of 194.2 μU/L/min at 2.5 μM BWA concentration, representing 19% activity enhancement. The PON-1 activity enhanced linearly by increasing BWA concentration from 0 to 0.5 μM and, after that, remained almost constant with the further enhancement of the BWA concentration, while compared to $CoQ_{10}$, BWA displayed a significantly 20% higher ($p = 0.041$) PON-1 activity at 2.5 μM concentration. Interestingly, the PON-1 activity in response to $CoQ_{10}$ was found to be nearly similar to the activity observed in the control (without BWA or $CoQ_{10}$), signifying no effect of $CoQ_{10}$ at the tested concentrations (0.01–2.5 μM) on the activity enhancement of PON-1. To the best of our knowledge, no study has documented the effect of BWA on the activity enhancement of the antioxidant enzyme PON-1. However, some previous studies document the impact of BWA on the activity enhancement of antioxidant enzymes such as catalase, superoxide dismutase, and glutathione peroxidase [19,37]. The outcome of the present study adds the value that BWA can enhance PON-1 activity and strengthen the use of BWA as a suitable nutraceutical precisely in relation to the biofunctionality of HDL.

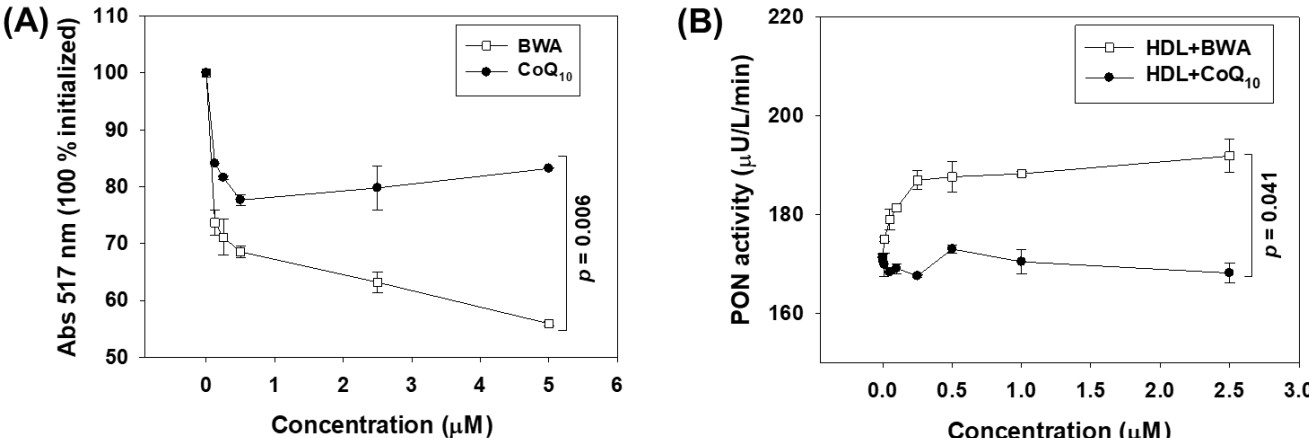

**Figure 2.** The comparative effect of beeswax alcohol (BWA) and coenzyme $Q_{10}$ ($CoQ_{10}$) on free radical scavenging and paraoxonase 1 (PON-1) activity. (**A**) DPPH free radical scavenging activity. (**B**) Paraoxonase 1 (PON-1) activity in HDL. The PON-1 activity is expressed as the formation of 1 μmol *p*-nitrophenol from the substrate paraoxon-ethyl at a unit volume of the enzyme and time. *p*-value represents the statistical difference between the groups employing one-way ANOVA following Tukey's post hoc analysis.

### 3.2. Inhibition of Cupric Ion ($Cu^{2+}$)-Mediated LDL Oxidation by BWA

LDL is vulnerable to oxidation, which leads to the accumulation of oxidative LDL (oxLDL) that is readily uptaken by the macrophages via CD36 and Sr-A scavenger receptors, rendering macrophages into foam cells and causing atherosclerotic lesions [38–40]. Knowing the several detrimental consequences of oxLDL, in the present study, the effect of BWA was evaluated in the inhibition of the oxidation of LDL induced by $CuSO_4$, which is considered an excellent external stressor owing to the ability of $Cu^{2+}$ ions to cause oxidative damage to LDL protein (apo-B) and lipid portion [26]. At the onset, the effect of BWA was evaluated in the inhibition of the $Cu^{2+}$ ion-mediated oxidation of LDL (apo-B) using native gel electrophoresis. As depicted in Figure 3A, the native LDL (lane N) showed the retardation of electrophoretic mobility, while the oxLDL (LDL + $CuSO_4$) (lane O) showed higher electrophoretic mobility. The treatment of BWA efficiently prevents $CuSO_4$-mediated oxidative damage of LDL in a dose-dependent manner (lanes 4–6). As documented in Figure 3A, the electrophoretic mobility of LDL progressively retarded in response to BWA (10–30 μM) treatment. Noticeably, at 30 μM concentration of BWA, the LDL electrophoretic mobility front (lane 6) is the same as the electrophoretic mobility front of native LDL (lane N), signifying the impact of BWA to prevent the oxidative damage of LDL. On the other side, $CoQ_{10}$ (10–30 μM) also displayed a dose-dependent effect to prevent LDL oxidation as evidenced by retardation of electrophoretic mobility (lane 1–3). Notably, compared to $CoQ_{10}$, BWA showed slightly better prevention of LDL oxidation at the tested concentrations (10–30 μM), as evidenced by the higher band intensity with more retarded electrophoretic mobility at the respective concentrations.

Furthermore, the extent of lipid oxidation in LDL, which was exposed to $CuSO_4$ and subsequently treated with BWA or $CoQ_{10}$, was evaluated. Results demonstrate maximum lipid oxidation (evident by the higher MDA quantification) in LDL treated with $CuSO_4$ (Figure 3B). A maximum of 91.2 μM MDA was quantified in LDL treated with $CuSO_4$, which is, significantly, 21-fold higher ($p = 0.001$) than the MDA level quantified in native LDL. The treatment of BWA effectively inhibits $CuSO_4$-mediated lipid oxidation in a dose-dependent manner. At 10, 20, and 30 μM BWA concentrations, a 1.9-fold ($p < 0.001$), 2.1-fold ($p < 0.001$), and 2-fold ($p < 0.001$) low LDL oxidation was observed as compared to the LDL treated with $CuSO_4$, signifying the role of BWA to prevent lipid oxidation of LDL. The preventive effect of BWA was further confirmed by performing $CuSO_4$-mediated LDL oxidation in the higher amount of BWA (0–100 μM). The results suggest a dose-dependent effect of BWA

to prevent the $Cu^{2+}$ ion-mediated oxidation of LDL (apo-B) and confirm the effective role of BWA in preventing LDL oxidation (Supplementary Figure S1). Surprisingly, the $CoQ_{10}$ displayed a nonsignificant effect on LDL lipid peroxidation altered by $CuSO_4$. The most probable reason for BWA to prevent LDL oxidation relates to its antioxidant activity, which scavenges harmful free radicals and ions, in turn creating a healing effect against oxidative stress. A prior study [32] highlighted the metal-chelating activity of octacosanol, thereby corroborating the current results, given that octacosanol is an essential component of BWA. Additionally, the results follow the previous reports suggesting the impact of BWA on the prevention of lipid peroxidation and oxidative damage of protein in the gastric mucosa of rats [18,19,41]. However, to the best of our knowledge, no study so far has documented the effect of BWA on preventing LDL protein (apo-B) and lipid peroxidation.

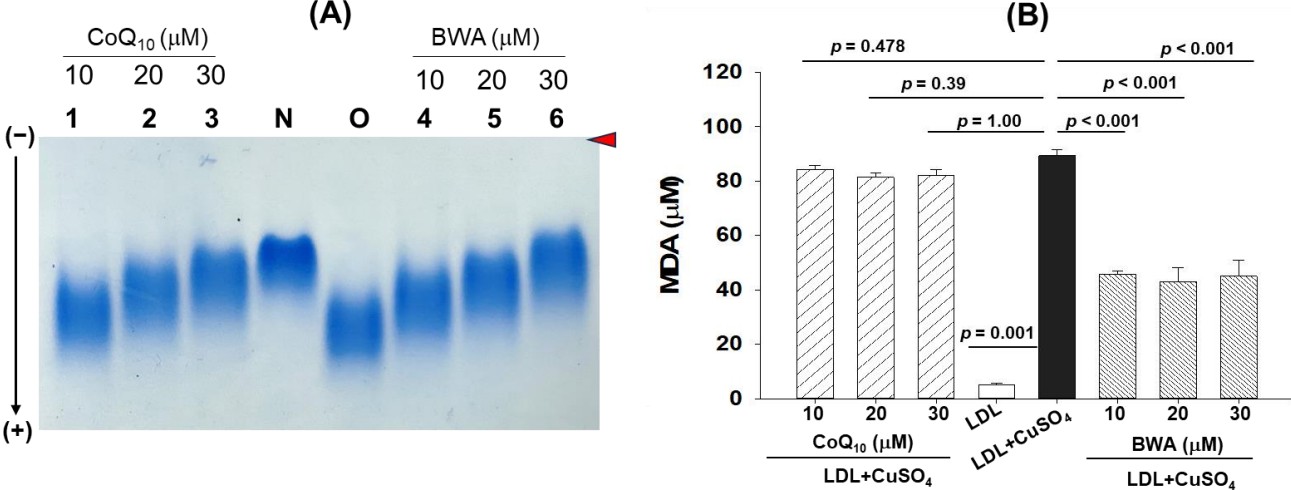

**Figure 3.** Effect of beeswax alcohol (BWA) and coenzyme $Q_{10}$ ($CoQ_{10}$) on $CuSO_4$-induced oxidation of LDL. (**A**) Electrophoretic mobility of apo-B fraction of LDL. Lanes 1, 2, and 3 represent LDL + $CuSO_4$ treated with 10, 20, and 30 μM $CoQ_{10}$, respectively. Lanes N and O reflect a native non-oxidized LDL and $CuSO_4$-oxidized LDL, respectively. Lanes 4, 5, and 6 symbolize LDL + $CuSO_4$ treated with 10, 20, and 30 μM BWA. Electrophoresis was performed on 0.5% agarose gel using Tris-EDTA buffer (pH 8.0) at a constant voltage (50 V). The red arrow indicates the gel loading front. (**B**) Quantification of $CuSO_4$-induced LDL oxidation in the presence and absence of BWA and $CoQ_{10}$. The LDL oxidation was quantified by thiobarbituric acid reactive substance (TBARS) assay using malondialdehyde (MDA) as reference. *p* value represents the statistical difference between the groups with respect to oxidized LDL employing one-way ANOVA following Tukey's post hoc analysis.

### 3.3. BWA Restored Cupric Ion ($Cu^{2+}$)-Mediated Morphological Changes in LDL

Transmission electron microscopy (TEM) is a useful technique to provide crucial insight into the structural characteristics of LDL particles [22]. As depicted in Figure 4, the native LDL appeared with a circular shape with even surface morphology without any particle aggregation, with the highest average particle size being $514.2 \pm 23.8$ nm$^2$. The exposure of $CuSO_4$ adversely impacts the typical LDL particle characteristic. As depicted in Figure 4, an irregular size, distorted surface morphology with notable particle aggregation, and reduced LDL particle size was observed in LDL treated with $CuSO_4$. The average particle size of native LDL ($514.2 \pm 23.8$ nm$^2$) decreased significantly by 40.8% ($p < 0.001$) in response to the exposure of $CuSO_4$, suggesting the adversity of $CuSO_4$ towards the morphological alteration of LDL. Conversely, when LDL was exposed to BWA at the concentration of 10–40 μM, a noteworthy reversal in the $CuSO_4$-mediated structural alteration was observed. At 10, 20, 30, and 40 μM BWA concentration, the LDL particle size significantly enhanced by 18.4% ($p = 0.014$), 29.45% ($p = 0.001$), 25.9% ($p = 0.001$), and 40.2% ($p < 0.001$), respectively, compared to the particle size appearing in $CuSO_4$-treated LDL. Moreover, the LDL particle morphology altered by $CuSO_4$ improved in response to BWA;

precisely at 40 μM concentration, LDL tends towards the normal morphology, as it appeared in the native LDL. The findings collectively suggest that BWA possesses a counteractive effect against the oxidative challenge posed by $CuSO_4$, thereby affording the protection of LDL particles from the detrimental effect of oxidative stress. Previously, several studies documented the antioxidant effect of BWA [12,42]; however, no study described the effect of BWA to prevent LDL morphological changes induced by oxidative stress.

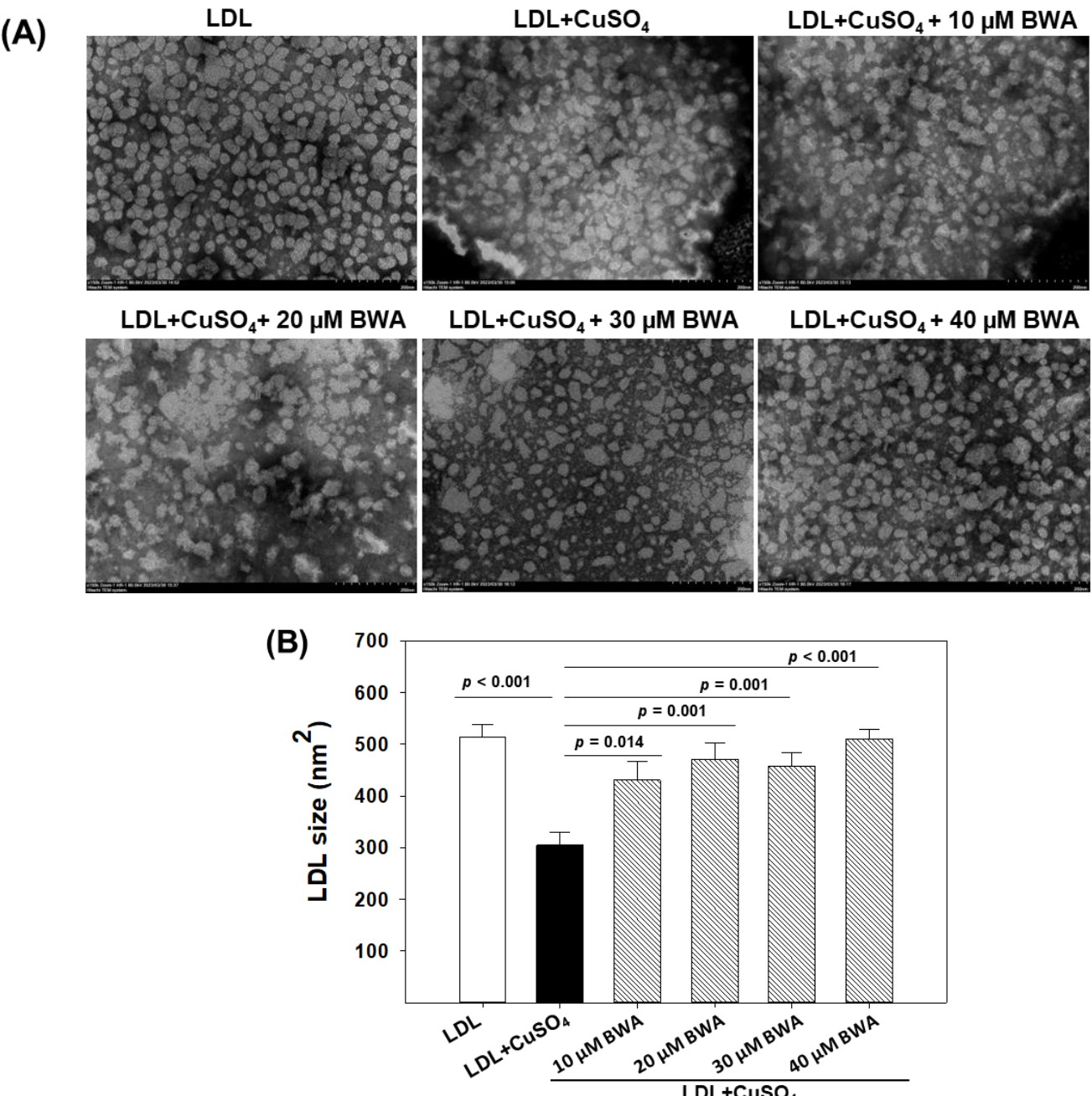

**Figure 4.** Effect of beeswax alcohol (BWA) on size and structural alteration of LDL probed by transmission electron microscope (TEM). (**A**) TEM images of LDL in the presence and absence of $CuSO_4$ and $CuSO_4$ + BWA (10–30 μM). Images were captured at 150 K magnification after negative staining with phosphotungstic acid (graphic scale = 200 nm). Supplementary Figures S2–S7 present magnified views of the images. (**B**) Quantification of LDL size in response to different treatments. *p*-value represents the statistical difference between the groups with respect to oxidized LDL employing one-way ANOVA following Tukey's post hoc analysis.

### 3.4. BWA Prevents HDL Degradation by Ferrous Ions (Fe²⁺)

HDL is the most important lipoprotein, with multifaceted functions including in cardiovascular health [36,43]. Notably, the important role of HDL in preventing LDL oxidation is a key feature to deter atherosclerotic lesions. Perceiving the imperative role of HDL in cardiovascular and other diseases, the impact of BWA was evaluated in the prevention of the degradation of HDL stimulated by $FeSO_4$. Figure 5 depicts a compelling visual representation of the effect of various treatments on HDL's (apoA-I) integrity at 72 h of treatment. The native HDL (lane 1) exhibited the baseline conditions with a sharp, intensified band. However, HDL exposed to $Fe^{2+}$ ions (final 60 μM) suffered severe degradation (lane 2), evident by a 1.3-fold lower band intensity compared to the band intensity of native HDL (lane 1). The cotreatment of HDL with BWA, specifically with BWA at 20 and 30 μM, effectively prevents the degradation of HDL provoked by $Fe^{2+}$ ions, evident in 1.3-fold and 1.4-fold higher band intensities (lanes 4 and 5), respectively, compared to HDL treated with $Fe^{2+}$ ions (lane 2).

The morphology of HDL in response to different treatments was evaluated by TEM, an efficient tool to assess the structural deformities of HDL [24]. It has been well established that external stress significantly affects HDL morphology and quality, eventually affecting HDL functionality [24]. The results depicted in Figure 5A,B revealed an even surface morphology with a bigger particle size ($142.3 \pm 11.2$ nm$^2$) of the native HDL that significantly reduced by 33.1% ($p = 0.007$) when HDL was exposed to $Fe^{2+}$ ions. Treatment of BWA, specifically 30 μM, efficiently reverted $Fe^{2+}$ ion-impaired structural changes. A $144.5 \pm 6.2$ nm$^2$ HDL particle size was observed in 30 μM BWA-treated HDL, i.e., significantly, 34.4% higher ($p = 0.002$) than the particle size observed in HDL treated with $Fe^{2+}$ ions, signifying the protective effect of BWA to counter the adversity posed by the exposure of $Fe^{2+}$ ions.

These results serve as compelling evidence to support the notion that BWA plays a crucial preventive role in mitigating the oxidative damage inflicted upon HDL by $Fe^{2+}$ ions. The protective function of BWA underscores its significance in maintaining the structural integrity and functional prowess of HDL impaired by oxidative challenge. So far, no study has elucidated the effect of BWA on preventing HDL degradation; however, a few studies have documented the effects of some biological materials to prevent HDL degradation against the challenge posed by $Fe^{2+}$ ions primarily by their antioxidant role [44] and thereby strengthening the present notion that BWA, owing to its antioxidant property, prevents HDL degradation.

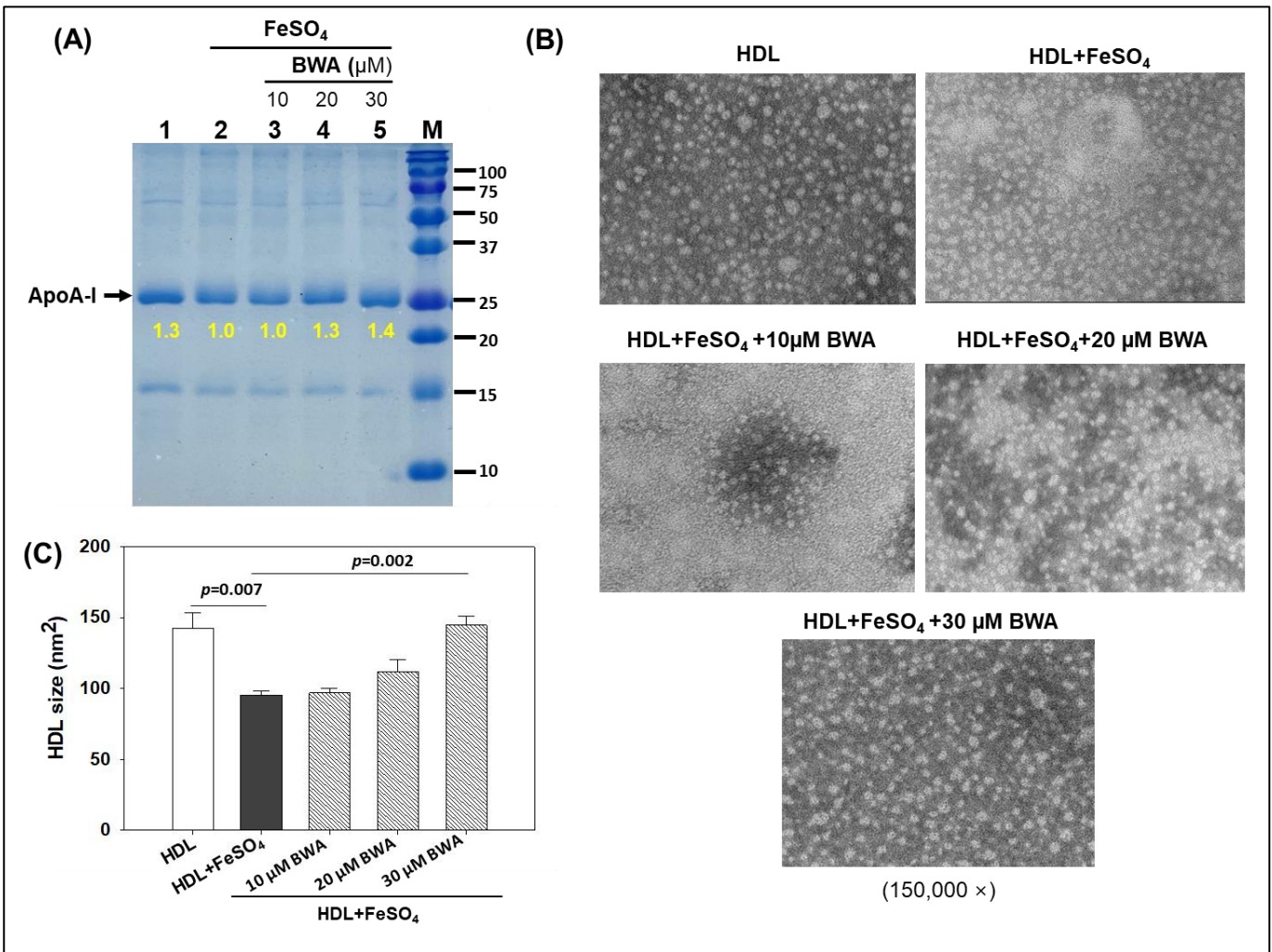

**Figure 5.** Effect of beeswax alcohol (BWA) on ferrous ion (Fe$^{2+}$)-mediated proteolytic degradation and structural alteration in HDL. (**A**) Electrophoretic mobility of apoA-I fractions of LDL. Samples were run in 15% SDS-PAGE after 72 h incubation and subsequently stained with Coomassie brilliant blue (0.125%). Lane 1 represents HDL alone. Lane 2 represents HDL + FeSO$_4$ (60 μM), while lanes 3, 4, and 5 contain HDL + FeSO$_4$ (60 μM) in the presence of 10, 20, and 30 μM BWA, respectively. Lane M molecular weight marker (10–250 kDa). Samples in lanes 1–5 were incubated at 72 h. (**B**) TEM images of HDL (150,000×) in the presence and absence of FeSO$_4$, FeSO$_4$ + BWA (10–30 μM). (**C**) Quantification of HDL size in response to different treatments. *p* value represents the statistical difference between the groups with respect to oxidized LDL employing one-way ANOVA following Tukey's post hoc analysis.

### 3.5. BWA Enhances the CML-Impaired Embryo Survivability

Due to their high genomic similarity with humans, zebrafish are considered an excellent animal model for preclinical studies [45,46]. Herein, the effect of BWA was evaluated in the protection of zebrafish embryos against carboxymethyllysine (CML) induced toxicity. CML is an important advanced end glycation (AGE) product that provokes oxidative stress and inflammation and has several detrimental effects [44,47]. As depicted in Figure 6A, the highest embryo survivability was observed in the PBS-alone group. In contrast, the embryos injected with CML showed a gradual decline in embryo survivability, reaching 47.3% at 5 h post-treatment. Conversely, the injection of BWA (15 μM) effectively enhances the CML-impaired embryo survivability. At 5 h post-treatment, 66.6% of embryo survivability was observed in the BWA + CML-injected group, which is, significantly, 1.4-fold higher (*p* = 0.041) than the embryo survivability in the group only injected with CML. Surprisingly,

no impact of $CoQ_{10}$ (15 μM) was observed on the CML-impaired embryo survivability; furthermore, much lower embryo survivability at 5 h post-treatment was observed in the $CoQ_{10}$ + CML-injected group compared to the only-CML-injected group.

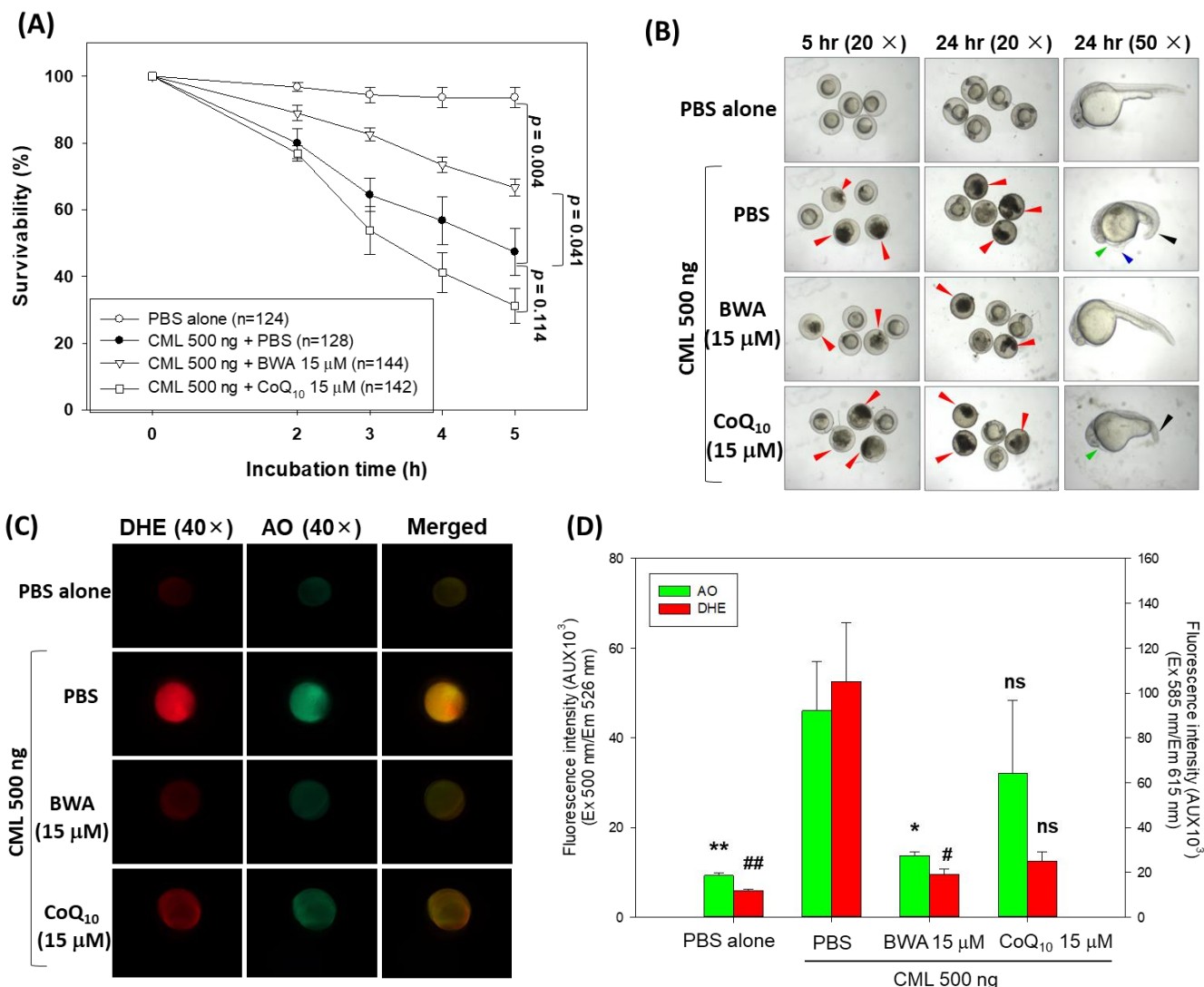

**Figure 6.** A comparative effect of beeswax alcohol (BWA) and coenzyme $Q_{10}$ ($CoQ_{10}$) to prevent carboxymethyllysine (CML)-induced toxicity in zebrafish embryos. (**A**) Survivability rate of zebrafish embryos at 5 h post-treatment. (**B**) Images representing developmental deformities at 5 h and 24 h post-treatment. The red and black arrows represent the embryo death and tail curvature, respectively, while the green and blue arrows represent pericardial and yolk sac edema, respectively. (**C**) Fluorescent images of dihydroethidium (DHE) and acridine orange (AO) representing the reactive oxygen species (ROS) production and apoptosis in the embryos of zebrafish. (**D**) Quantification of the fluorescent intensity. Image J software was utilized to compute the fluorescent intensity of DHE (red fluorescence) and AO (green fluorescence) staining. PBS and CML groups represent the embryos microinjected with 10 nL of PBS and 500 ng CML/10 nL PBS, respectively, while embryos in CML + BWA or $CoQ_{10}$ groups were microinjected with 500 ng CML suspended in 10 nL of BWA or $CoQ_{10}$ (final 15 μM). ** represents $p < 0.01$, * represents $p < 0.05$ compared with the AO fluorescent intensity of only-CML-injected group, while ## represents $p < 0.01$, and # represents $p < 0.05$ compared with the DHE fluorescent intensity of only-CML-injected group; "ns" represents a nonsignificant difference between the groups.

Morphological changes in the developing embryos were examined at 5 and 24 h post-treatment (Figure 6B). A severe embryo death (indicated by the red arrow) was observed in the CML-injected group; in contrast to this, no embryo death was observed in the PBS-injected group. A delayed embryo development with several developmental deformities concerning tail fin curvature (indicated by the black arrow), yolk sac edema (indicated by the blue arrow), and pericardial edema (indicated by the green arrow) was observed in embryos injected with CML. In the embryos co-injected with BWA, we documented the prevention of CML-induced developmental impairment, and this observation closely correlated with the survivability outcomes depicted in Figure 6A. In contrast, the non-plausible effect of $CoQ_{10}$ was observed in embryo development altered by the CML. Notably, a curvature of the tail (indicated by the black arrow) and pericardial edema (indicated by the green arrow) were observed in the embryos injected with $CoQ_{10}$ + CML. The findings demonstrated the protective effect of BWA against CML-induced developmental abnormalities and mortality in zebrafish embryos.

The fluorescent images of DHE and AO staining document the extent of ROS production and apoptosis induced by CML (Figure 6C,D). The results of DHE staining unveiled a massive ROS production in response to CML, which is, significantly, 9.5-fold higher ($p = 0.024$) than the ROS level in the PBS-injected embryos. The treatment of BWA efficiently thwarts CML-induced ROS production. A significant 6-fold lower ($p = 0.031$) DHE fluorescent intensity was quantified in BWA + CML-injected embryos compared to only-CML-injected embryos. The AO staining revealed the highest fluorescent intensity in the CML-injected group, which was significantly 5.1-fold higher ($p = 0.009$) than the fluorescent intensity observed in the PBS-injected group, implying a high extent of apoptosis. BWA significantly prevents CML-induced apoptosis, as evidenced by, significantly, 3.4-fold lower ($p = 0.014$) AO fluorescent intensity compared to the CML-injected embryos. In contrast, no preventive effect of $CoQ_{10}$ was observed against CML-induced apoptosis.

The findings of DHE and AO staining collectively endorse the efficacy of BWA in preventing CML-induced ROS production and apoptosis. The antioxidant nature of BWA is a major factor that efficiently counters the CML-induced ROS and consequently improves embryo survivability. Our results agree with previous reports suggesting that the antioxidants can revert the CML-induced ROS production and apoptosis, consequently rescuing the zebrafish embryos from CML-induced toxicity.

### 3.6. Consumption of BWA Improved Oxidative Plasma Variables in Human Subjects

The in vitro and preclinical studies performed in the zebrafish suggested the antioxidant properties of BWA. To expand upon the findings observed in vitro, we conducted a preliminary clinical study to assess the effects of consuming BWA (100 mg/day) for 12 weeks on oxidative plasma variables in middle-aged and older individuals. The choice of participants encompassed middle-aged and older individuals due to the well-established correlation between aging and the reduction in the inherent antioxidant defense system, resulting in an elevated level of oxidative stress associated with various age-related diseases [48,49]. Initially, the MDA levels were evaluated, which are considered an important marker of lipid peroxidation. The results demonstrated a significant 25% reduction ($p < 0.001$) in MDA levels after 12 weeks of consumption of BWA compared to the baseline values (Figure 7A and Table 1). In contrast, no significant changes in MDA levels were noticed after the 12-week consumption in the placebo group compared to their baseline values.

Further, total hydroperoxide levels were evaluated in both BWA and placebo groups. Hydroperoxides are formed by the peroxidation of unsaturated fatty acids and cholesterol [31] and are a good indicator of oxidative stress [50,51]. The results document a significant 27% reduction ($p < 0.001$) in plasma total hydroperoxides in the BWA-consuming group compared to their baseline values. In contrast, nonsignificant changes in serum total hydroperoxides levels were observed in the placebo group, signifying the effect of BWA on the reduction in total hydroperoxides (Figure 7B and Table 1). Simultaneously, a significant

elevation of 22.2% in plasma total antioxidant status was noticed after 12 weeks of BWA consumption compared to the baseline value. Contrary to this, the placebo group's total antioxidant status remains unchanged (Figure 7C and Table 1).

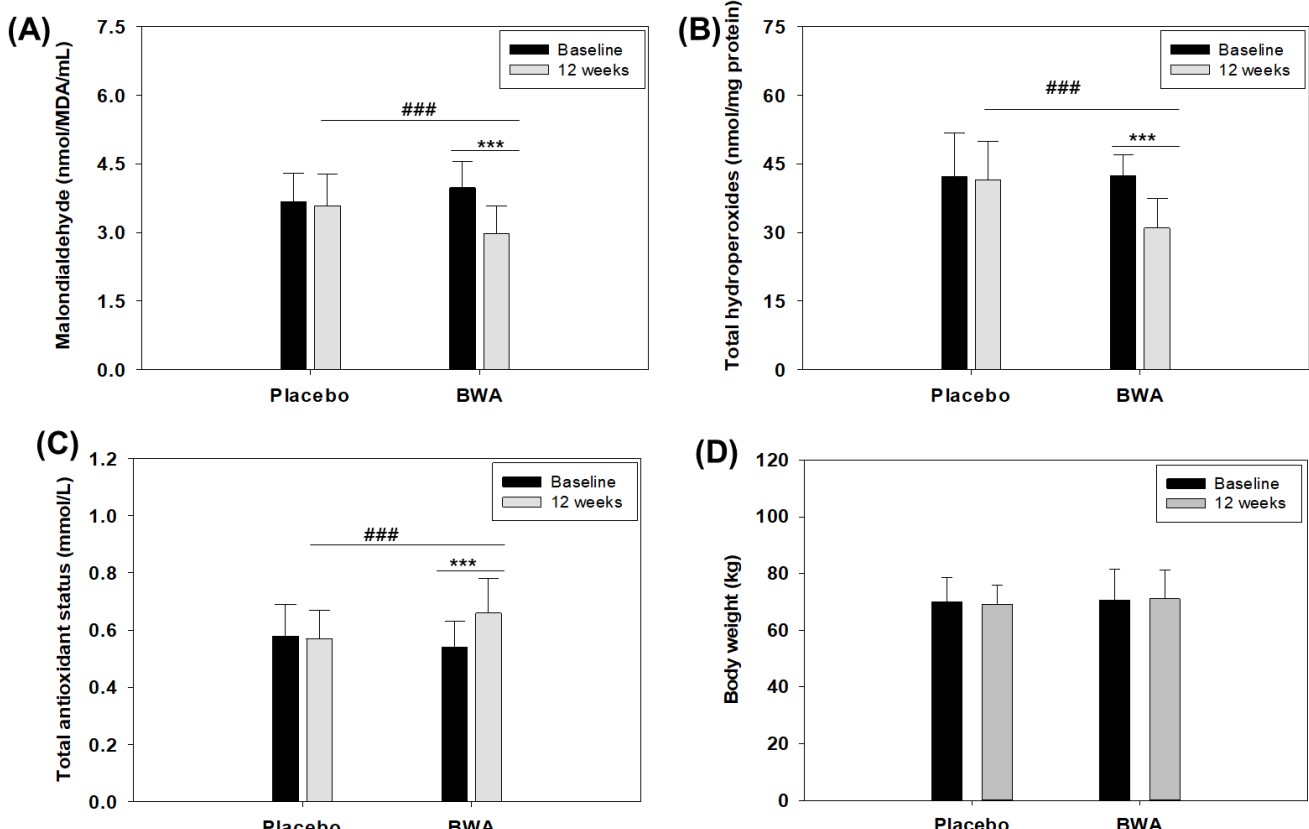

**Figure 7.** Oxidative plasma variables and body weight of humans (*n* = 50) before and after 12 weeks of intake of beeswax alcohol (BWA) (100 mg/day) and placebo. (**A**) Malondialdehyde (MDA). (**B**) Total hydroperoxides. (**C**) Total antioxidant status. (**D**) Body weight. *** represents the statistical difference between the 12-week intake of BWA with its baseline values, while ### represents a statistical difference between the groups' 12-week intake of BWA and placebo.

The collective outcomes from the plasma oxidative variables illustrate the efficacy of BWA in mitigating oxidative stress in middle-aged and elderly individuals. These findings align well with previous reports highlighting the impact of BWA on lipid peroxidation [52] and oxidative markers [53]. Likewise, another report deciphered the impact of BWA on the reduction of hydroxyl radical generation, lipid peroxidation, and activity enhancement of antioxidant enzymes in the gastric mucosa of rats [12,54,55]. Furthermore, the diminished oxidative plasma variables observed in groups consuming BWA might be linked to the augmentation of PON-1 activity, an important antioxidant associated with HDL [35,36] that prevents lipid peroxidation in LDL [56]. The notion is supported by our in vitro findings that highlight the influence of BWA on improving PON-1 activity. Similarly, there are reports highlighting BWA's influence on the production of hydroxyl radicals, protein oxidation, and the augmentation of antioxidant enzyme activities like catalase, superoxide dismutase (SOD), and glutathione peroxidase (GSH-Px) [19,41], as well as plasma total antioxidant status [42,52,57], reinforcing the current finding.

*3.7. BWA Has No Adverse Effect on Anthropometric Profile, Liver and Kidney Biomarkers, and Lipid Profile*

To assess the suitability of BWA for human consumption, it is imperative to evaluate its safety. No discernible impact of BWA was observed on the anthropometric profile

concerning body weight (Figure 7D), pulse rate (beat/min), and blood pressure during 12 weeks of consumption (Table 1). As depicted in Table 1, the anthropometric profile remained unchanged in the placebo and BWA-supplemented group during 12 weeks compared to baseline values. Furthermore, the blood analysis revealed no adverse effect of BWA consumption on glucose, hemoglobin, hematocrit, red blood cell count, white blood cell, and total platelets count, along with aspartate amine transferase (AST), alanine amine transferase (ALT), and creatinine levels (Table 1). Also, BWA consumption did not affect total cholesterol and triglyceride levels throughout this study, as evidenced by the unchanged serum levels of total cholesterol and triglycerides from the baseline values (Table 1). Notably, a minor adverse effect was observed in five subjects. Among them, one belonged to the BWA group and reported mild diarrhea, while the remaining four were from the placebo group, comprising two incidences of mild diarrhea and two incidences of mild urinary problems. The findings collectively affirm the safety profile of BWA and its favorable tolerability in the context of this study.

**Table 1.** Effect of 12 weeks' intake of beeswax alcohol (BWA) (100 mg/day) and placebo on anthropometric profile, blood analysis, liver and kidney function biomarkers, and lipid profile.

| Treatment | Baseline (Week 0) | Week 12 |
|---|---|---|
| Malondialdehyde (nmol/MDA/mL) | | |
| Placebo | $3.67 \pm 0.63$ | $3.59 \pm 0.68$ |
| BWA | $3.98 \pm 0.58$ | $2.98 \pm 0.61$ ***,### |
| Total hydroperoxides (nmol/mg protein) | | |
| Placebo | $42.31 \pm 9.37$ | $41.57 \pm 8.39$ |
| BWA | $42.43 \pm 4.49$ | $30.97 \pm 6.56$ ***,### |
| Total antioxidant status (mmol/L) | | |
| Placebo | $0.58 \pm 0.11$ | $0.57 \pm 0.10$ |
| BWA | $0.54 \pm 0.09$ | $0.66 \pm 0.12$ ***,### |
| Pulse (beats/min) | | |
| Placebo | $70.08 \pm 3.34$ | $70.25 \pm 1.07$ |
| BWA | $71.36 \pm 3.09$ | $70.56 \pm 1.47$ |
| Diastolic blood pressure (mmHg) | | |
| Placebo | $79.20 \pm 4.93$ | $78.33 \pm 3.61$ |
| BWA | $78.80 \pm 4.40$ | $78.40 \pm 3.74$ |
| Systolic blood pressure (mmHg) | | |
| Placebo | $123.40 \pm 11.06$ | $125.20 \pm 5.41$ |
| BWA | $125.60 \pm 11.21$ | $125.00 \pm 6.12$ |
| Hemoglobin (g/L) | | |
| Placebo | $12.91 \pm 1.31$ | $12.98 \pm 1.15$ |
| BWA | $12.77 \pm 0.83$ | $12.80 \pm 0.85$ |
| Hematocrit (%) | | |
| Placebo | $39.31 \pm 2.64$ | $39.46 \pm 2.06$ |
| BWA | $38.98 \pm 2.06$ | $39.50 \pm 1.81$ |
| Red blood cells count ($\times 10^{12}$/L) | | |
| Placebo | $4.23 \pm 0.38$ | $4.17 \pm 0.20$ |
| BWA | $4.22 \pm 0.33$ | $4.26 \pm 0.28$ |
| White blood cells count ($\times 10^{9}$/L) | | |
| Placebo | $6.00 \pm 1.43$ | $6.09 \pm 1.15$ |
| BWA | $6.38 \pm 1.34$ | $6.15 \pm 0.98$ |

**Table 1.** *Cont.*

| Treatment | Baseline (Week 0) | Week 12 |
|---|---|---|
| | Platelets count ($\times 10^9$/L) | |
| Placebo | 225.70 ± 39.33 | 227.41 ± 35.56 |
| BWA | 213.84 ± 42.42 | 215.36 ± 35.78 |
| | Aspartate aminotransferase (U/L) | |
| Placebo | 24.80 ± 6.26 | 24.75 ± 3.82 |
| BWA | 26.28 ± 5.42 | 25.60 ± 4.23 |
| | Alanine aminotransferase (U/L) | |
| Placebo | 16.88 ± 5.59 | 17.58 ± 4.57 |
| BWA | 17.88 ± 7.13 | 17.32 ± 4.23 |
| | Glucose (mmol/L) | |
| Placebo | 4.36 ± 0.90 | 4.39 ± 0.72 |
| BWA | 4.92 ± 1.05 | 4.61 ± 0.93 |
| | Creatinine (μmol/L) | |
| Placebo | 76.08 ± 16.91 | 77.28 ± 12.10 |
| BWA | 76.12 ± 20.29 | 77.67 ± 14.33 |
| | Total cholesterol (mmol/L) | |
| Placebo | 6.31 ± 1.37 | 6.11 ± 1.10 |
| BWA | 6.34 ± 1.12 | 6.08 ± 0.78 |
| | Triglycerides (mmol/L) | |
| Placebo | 1.95 ± 0.98 | 1.71 ± 0.53 |
| BWA | 1.78 ± 0.74 | 1.68 ± 0.49 |

BWA refers to beeswax alcohol. *** represents the statistical difference ($p < 0.001$) between the 12 weeks intake of BWA with its baseline values, while ### represents a statistical difference ($p < 0.001$) between the group's 12 weeks intake of BWA and placebo.

### 3.8. Limitations

The present study's outcome (preclinical and clinical) endorses BWA as a safe supplement to prevent oxidative stress and related disorders; however, a small number of participants in the clinical trials emerged as the basic limitation of this study, which will be addressed in the future by conducting clinical trials with a larger participant pool. Also, the effect of BWA consumption on HDL-C and LDL-C levels and HDL-associated PON-1 activity in the clinical samples will be examined to strengthen the current findings. The consumption of a low BWA dose (50 mg/day) will be included in the future study to explore a comparative dose-dependent effect of BWA on the oxidative variables.

## 4. Conclusions

In vitro, BWA demonstrated substantial antioxidant properties and enhanced PON-1 activity within HDL. BWA's antioxidative effect efficiently thwarted LDL oxidation and significantly preserved the integrity and structural stability of HDL/apoA-I. Also, BWA was shown to prevent CML-induced oxidative stress and apoptosis, consequently rescuing zebrafish embryos from developmental deformities and mortality (Figure 8). A preliminary clinical investigation involving non-dyslipidemic middle-aged and older individuals confirmed the safety of BWA and its efficacy in reducing oxidative stress variables and elevating total antioxidant status. Nonetheless, the dual functionality of BWA in activating PON-1 within HDL and inhibiting LDL oxidation indicates the efficacy of BWA against cardiovascular disorders that require a comprehensive investigation in the future.

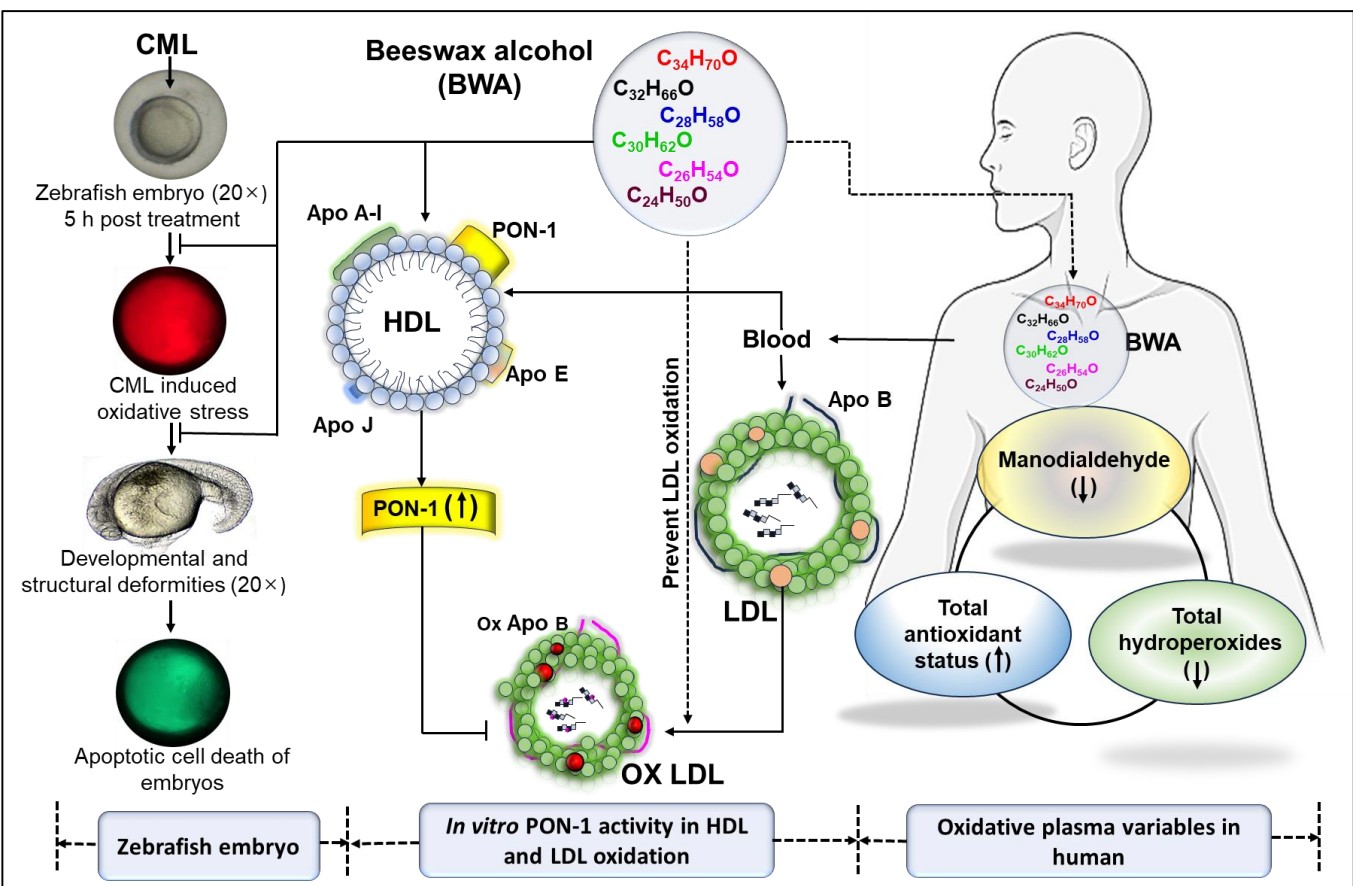

**Figure 8.** Summary of antioxidant effect of beeswax alcohol (BWA) to prevent oxidative damage in zebrafish embryos, low-density lipoproteins (LDL), enhancement of paraoxonase 1 (PON-1) activity of HDL, and improvement of human oxidative plasma variable.

**Supplementary Materials:** The following supporting information can be downloaded at: https://www.mdpi.com/article/10.3390/cimb46010026/s1, Figure S1: Effect of beeswax alcohol (BWA) (0–100 μM) on CuSO₄-induced oxidation of LDL. (A) Electrophoretic mobility of apo-B fraction of LDL. (B) Quantification of CuSO4-induced LDL oxidation in the presence and absence of BWA. Figures S2–S7: Magnified image representing the effect of beeswax alcohol (BWA) on size and structural alteration of LDL probed by transmission electron microscope (TEM). Figure S2: TEM images of native LDL. Figure S3: TEM images of LDL treated with CuSO₄. Figures S4–S7: LDL treated with CuSO₄ in the presence of 10, 20, 30, 40 BWA μM BWA, respectively.

**Author Contributions:** Conceptualization, K.-H.C.; methodology, S.-H.B., H.-S.N., A.B., L.E.L.-G., I.R.-C., J.I.-F., J.C.F.-T., V.M.-C., Y.P.-G., A.O.Y. and S.M.-C.; data curation, writing—original draft preparation, K.-H.C. and J.C.F.-T.; writing—review and editing, K.-H.C.; supervision, K.-H.C.; All authors have read and agreed to the published version of the manuscript.

**Funding:** This research received no external funding.

**Institutional Review Board Statement:** The animal study protocol was approved by the Committee of Animal Care and Use of Raydel Research Institute (approval code RRI-20-004). The clinical study was approved by the Clinical Research Ethics Committee of the Surgical Medical Research Center (IRB approval number. 04-19), Havana, Cuba, and registered in the Public Registry of Clinical Trials of Cuba (RPCEC00000401).

**Informed Consent Statement:** Not applicable.

**Data Availability Statement:** The data used to support the findings of this study are available from the corresponding author upon reasonable request.

**Conflicts of Interest:** The authors declare no conflicts of interest.

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
