# Peer review of "Beeswax Alcohol Prevents Low-Density Lipoprotein Oxidation and Demonstrates Antioxidant Activities in Zebrafish Embryos and Human Subjects: A Clinical Study"

_cimb, doi:10.3390/cimb46010026_

Round 1

Reviewer 1 Report

Comments and Suggestions for Authors

Page 1, line 31: include In vitro

Page 3, line 116: LDL and HDL in human blood: describe where this blood comes from.

page 7, line 276: correct this sentence  to Figure 2B

page 10, line 379 e 381: Remove CoQ10 from legend. These data are not in Figure 4A

page 12, line 424, figure 5: there is no lane 0

page 17, line 556: include In vitro

page 17, line 561: include non-dyslipidemic

 - Include plasma LDL-C and HDL-C concentrations in Table 1.

 -Include measurement of paraoxonase activity in the serum of Clinical Study participants. The article would be greatly strengthened by the inclusion of measurement of paraoxonase activity in serum, as this could corroborate with in vitro assays that showed increased PON-1 activity in the presence of BWA.

MacKness et al. showed that the HDL-associated enzyme, paraoxonase (PON1), is, at least in part, responsible for HDL's ability to prevent the accumulation of lipid peroxides in LDL. Discuss the reduction of oxidative stress in HDL, since this lipoprotein has antioxidant activities.

Mackness MI, Durrington PN, Arrol S, Evans AE, McMaster D, Ferrières J, Ruidavets JB, Williams NR, Howard AN. Paraoxonase activity in two healthy populations with differing rates of coronary heart disease. Eur J Clin Invest. 2000 Jan;30(1):4-10. doi: 10.1046/j.1365-2362.2000.00580.x. PMID: 10619995

Beeswax alcohol is extracted from beeswax and its composition depends on the flowers used by bees, the environmental climate variability, and the geographic location of the country in addition to good beekeeping practices.

How can the composition and quality of honey influence beeswax alcohol?

Could these variations of this product,  influence the results of future clinical studies?

This issue should be discussed as a limitation of the study.

Author Response

Thank you for your valuable comments and suggestions. 

Please find attached doc as point-to-point response.

Reviewer 2 Report

Comments and Suggestions for Authors

The study presented for review is practical and scientifically applied in nature. The authors used a large number of methodological approaches to prove the antioxidant effect of beeswax alcohol. They used a relatively new experimental model of zebrafish embryos as well as healthy volunteers. The manuscript is written at an excellent professional level with a very good command of the English and scientific language. There are still some small technical inconsistencies, such as in line 19 - significantly better them have to be significantly better than,  line 47 -......of the endogenous antioxidant’s leads, have to be of the endogenous antioxidants and leads to.....

It would be good, and for the benefit of the manuscript, to discuss the problems related to the gastrointestinal absorption of compounds and products of natural origin.

Comments on the Quality of English Language

Excellent scientific English language! 

Author Response

Thank you for your valuable comments, appreciations, and suggestions. 

Please find attached doc as point-to-point response.

Reviewer 3 Report

Comments and Suggestions for Authors

This is a very important and interesting study. This study is well designed, well planned research work. 
I have following suggestions to improve the manuscript. 
Please shorten the title. The manuscript title should be concise and concrete. 
Please improve English. 
Some repeatation is not required. For example, ImageJ was used for data analysis should be repeated wherever ImageJ was used, but repeating it's link is unnecessary. 
Please include a paragraph of limitations of the study if you think there were some limitations. 

Comments on the Quality of English Language

Please improve the quality of English allover the manuscript. 

Author Response

(The authors gave the same response as above.)

Reviewer 4 Report

Comments and Suggestions for Authors

Comments for authors: 

Title: 

Beeswax alcohol (BWA) protects lipoproteins from oxidation to rescue embryo death from acute toxicity and to exert antioxidant activities in serum of subjects with sixties individuals after 12 weeks consumption of BWA 100 mg/day

Authors: 

Cho K.-H., Fernandez-Travieso J. C., Bahuguna A., Molina-Cuevas V., Perez-Guerra Y., Yera A.O., Mendoza-Castano S.

Manuscript ID: cimb-2730485

Objective: 

In the present manuscript, the authors describe the effects of beeswax alcohol (BWA) as an antioxidant, preventing oxidative stress and the beneficial effects on LDL and HDL inclusive PON-1 activity, preventing deformities of zebrafish embryos induced by carboxymethyl lysine to subsume all these defending actions with a strength of evidence in case of a randomized clinical trial.

Points of criticism:

Title:

The title refers to embryos, but the species (zebrafish) is not mentioned. The subsequent designation of serum could give the erroneous impression that these are human embryos. In this regard, a clear assignment to zebrafish embryos should be included.

How was the dose determined, and is there a known LD50?

Page 3 – line 97:

Australia instead of Austrailia

Page 3 – line 104:

Coenzyme (Q10 CoQ10) should read Coenzyme Q10 (CoQ10)

Page 3 – Lines 123-132:

Why is a more extensive line spacing used in this area?

Page 4 – Lines 160 & 186:

Is there a particular reason that the link to the Image J software is underlined?

Abbreviations need to be explained at their first mention, e.g., AO Page 4 – line 182 (acridine orange).

The authors refer on page 7 - line 276 to an increase in PON-1 activity in Figure 2A - however, the PON-1 activity is shown in Figure 2B. This should be adjusted accordingly.

Figures 3, 5 and 6 have a black background, so some partial graphics are cut off or cannot be assigned due to missing labelling. It is, therefore, impossible to follow the explanations in the text. It is therefore urgently necessary to improve these figures significantly.

HDL was abbreviated as HLD several times, e.g., page 11 - line 394 - this needs to be improved.

Page 11 – line 407:

The authors write about the HDL size after treatment with 30 µM HDL. Presumably, the authors mean 30 µM BWA-treated HDL. This should be improved accordingly for readability and clarity.

Page 13 – lines 457-458:

The authors point out that BWA prevents abnormalities and mortality in zebrafish embryos. In Fig. 6 B, red arrows indicate tail curvature. This should be changed accordingly in the text.

Page 14 – line 508:

            The authors refer to hydroperoxides as important ROS. This term is incorrect, as peroxides do not have a radical character and are, therefore, not ROS per se.

Page 16 – line 546:

The authors point out that BWA does not influence total cholesterol or triglycerides. In line 546, they write that the “value” remains unchanged. Do they mean total cholesterol and triglyceride levels?

References:

In reference 21, there is a spelling mistake in "Size" (line 636) - this should be corrected.

Author Response

(The authors gave the same response as above.)

Round 2

Reviewer 1 Report

Comments and Suggestions for Authors

no comments

Author Response

Thank you for your valuable comments and suggestions. 

Thank you very much for acceptance

Reviewer 4 Report

Comments and Suggestions for Authors

Congratulations on this study, which is impressive in its informative value due to its elaborate experiments, including a clinical trial. Thank you for the professional review process, which has made the manuscript more meaningful.  

Author Response

(The authors gave the same response as above.)
